# Polarizabilities of Atoms in Molecules: Choice of the Partitioning Scheme and Applications for Secondary Interactions

**DOI:** 10.3390/molecules30204137

**Published:** 2025-10-20

**Authors:** Piero Macchi

**Affiliations:** Department of Chemistry, Materials, and Chemical Engineering, Politecnico di Milano, Via Mancinelli 7, 20131 Milano, Italy; piero.macchi@polimi.it

**Keywords:** polarizabilities, quantum chemical calculations, electron density, partitioning schemes, secondary interactions

## Abstract

This paper reviews methods for partitioning the molecular polarizability into atomic terms. The advantages and disadvantages of hard space or fuzzy partitioning are critically assessed and compared. Polarizability density is proposed as a key function for the in-depth analysis and prediction of molecular recognition and chemical reactivity. Applications in the study of secondary interactions are illustrated, particularly in electron donor–acceptor complexes and other molecular adducts.

## 1. Introduction

In chemistry, the distribution and movement of electrons in a system are two fundamental quantities, as first recognized by Schrödinger [1,2]. This implies that the electron charge density $\rho^{el}(r)$ and the electron current density $J^{el}(r)$ play a central role. Indeed, a careful analysis $\rho^{el}(r)$ (hereinafter, simply ρ(r)) informs on the occurrence and the nature of chemical bonding [3]. The interest of chemists, though, focuses mainly on the redistributions of electron density in molecules or solids that lead to supramolecular interactions, molecular recognition, and eventually chemical reactions. The latter are historically classified as *charge controlled* (hard-type) and *orbital controlled* (soft-type) [4]. The former depends on the uneven distribution of charge in a molecule, and the latter, instead, on the ability of a molecule to redistribute the charge, permanently or instantaneously. The orbitals involved in soft-type interactions are especially the *frontier* molecular orbitals, because they are more sensitive to an external perturbation.

Within the framework of McWeeney’s description of intermolecular interactions [5], the distinction between charge– and orbital–control means having larger interaction between, respectively, the ground states of the molecules (for which the electrostatic interaction energy is dominant) or between the ground state of one and excited states of the other (the polarization interaction energy dominates). Additionally, the interaction between excited states on both molecules implies another soft type of interaction, namely dispersion, introduced by London [6]. In this context, two quantities derived from $\rho^{el}(r)$ are important: (a) the electrostatic potential ϕ(r) (for the charge control) and (b) the polarizability density χ(r) (for the orbital control). While the former is often investigated and a huge array of studies have appeared based on the precise mapping of molecular ϕ(r) and its analysis [7], including experimental determinations [8], χ(r) is much less investigated. The polarizability density addresses the ability of a molecule to modify the electron density at each point in space.

Apart from the molecular recognition processes, many material properties depend on the polarizability, like opto-electronic and dielectric properties. Therefore, in-depth knowledge of the machinery of molecular polarizability is important not only for supramolecular chemistry but also for structure/property correlation and material design. In this respect, it is vital to understand how atoms and bonds between them shape the polarizability density and consequently the molecular polarizability. This fundamental task can be undertaken by partitioning the molecular polarizability into atomic terms and inspecting the transferability of the extracted quantities. Of note, there is no unique way to partition the molecular electron density into atoms, and the same holds true for the molecular polarizability.

This work has two purposes. One is comparing the atomic polarizabilities obtained with overlapping or non-overlapping atomic partitioning schemes. The second, more ambitious, aim is to propose the polarizability density and distributed atomic polarizabilities as tools for predicting the occurrence and the strength of secondary interactions.

## 2. Theoretical Background

As introduced above, the most important quantities for the evaluation of the intermolecular interactions are the electric potential and the polarizability density.

The electric potential is easily obtained from the total charge density distribution:(1)$$\phi \left(r\right)=\int \frac{\rho^{tot}\left(r\right)}{\left|r-r^{\prime }\right|}dr^{\prime }$$
where $\rho^{tot}(r)$ is the sum of electron and nuclear charge distributions $\rho^{N}\left(r\right)$. The latter is a discrete function, within the Born–Oppenheimer approximation, with the positive charge occurring only at the nuclear positions, whereas the distribution of negative charges is continuous in space. From the electric potential, one can derive the total electric charge at a point, using Poisson’s equation (here, expressed in atomic units):(2)$$\nabla^{2}\phi \left(r\right)=-\rho^{tot}\left(r\right)$$

For this reason, in the case of hard-type interactions, a good indicator for *nucleophilic* and *electrophilic* sites of a molecule comes from the inspection of the molecular electrostatic potential and even more of the electric field vector, $F=-\nabla \phi \left(r\right)$, which highlights the anisotropy of the interaction. While a quantitative mapping of F would return a more accurate quantification of hard-type interactions, its analysis would be more complicated, and it is not normally adopted. Instead, chemists prefer to reduce the information in terms of atom-based point charge distribution. For example, the most classical of secondary interactions, namely the hydrogen bond (HB), is usually described as the result of the electrostatic attractions between the positively charged proton of the HB donor group and the negatively charged HB acceptor group [9]. This is of course very concise information that explains, albeit approximately, many hard-type intermolecular interactions. Similar reasoning is adopted for the interpretation of other kinds of secondary interactions, often classified as hard type.

Soft interactions, instead, are governed by the ability of a molecule to re-polarize under the permanent or temporary electric field F generated by vicinal molecules. In this case, the polarizability density χ(r) best addresses the machinery. χ(r) is the field derivative of the total charge density distribution, i.e., a second-order tensor function consisting of the following elements:(3)$$\chi_{ij}\left(r\right)=r_{i}\frac{\partial \rho^{tot}\left(r\right)}{\partial F_{j}(r)}$$
where $r_{i}$ is the *i*-th component of the position vector r and $F_{j}(r)$ is the *j*-th component of the electric field vector at position r. While ρ(r) and $\phi \left(r\right)$ are scalar functions, hence easily visualized in 3D, χ(r) is a (dimensionless) tensor function of position, meaning that at each point r, a 3 × 3 matrix consisting of elements $\chi_{ij}\left(r\right)$ is defined. One can reduce the information of the full tensor using its trace $Tr\left[\chi (r)\right]$. This scalar function addresses the regions of a molecule that are more polarized but does not indicate along which direction and which component of the external electric field the molecular electron density is more perturbed. $Tr\left[\chi (r)\right]$ plays, for soft interactions, the same role as $\phi \left(r\right)$ for the hard ones, addressing electronegative and electropositive regions in a molecule, without indicating the directions along which the interaction would be more effective.

In Figure 1, $Tr\left[\chi (r)\right]$ is reported for some molecules investigated in this study. Some features of χ(r) and $Tr\left[\chi (r)\right]$ are interesting to comment on:The regions that are more polarizable are in the valence shell, which is quite clear because core electrons are more tightly bound and therefore more difficult to polarize.

$Tr\left[\chi (r)\right]$ is not positive everywhere. As can be seen in Figure 1, some regions in a molecule may polarize oppositely with respect to the applied electric field, as a reaction against stronger, direct polarization occurring in another part of the molecule. In fact, only the trace of the molecular polarizability α (obtained as space integration χ(r), see Equation (4)) is necessarily positive.

χ(r) is inherently not symmetric ($\chi_{ij}\left(r\right)\neq \chi_{ji}\left(r\right)$ for i≠j) unless the point r lies on a symmetry element of the molecule imposing some constraints on the tensor. Again, it is only the integrated quantity α, which is necessarily symmetric.

The molecular polarizability α is obtained by integrating χ(r) over the entire molecular volume. In fact, each component $\alpha_{ij}$ of the polarizability tensor is given by(4)$$\alpha_{ij}=\int \chi_{ij}\left(r\right)d^{3}r$$
which is equivalent to(5)$$\alpha_{ij}=\frac{\partial \mu_{i}}{\partial F_{j}}$$
where $\mu_{i}$ and $F_{j}$ are the i-th and j-th components of the molecular dipole moment and of a static and homogenous external electric field, respectively. The overall positivity and symmetry of the α tensor come from the fact that Equation (5) can also be written as a mixed double derivative of the molecular energy, which is of course insensitive to the permutation of the derivatives (because $\mu_{i}=\frac{\partial E}{\partial F_{i}}$, $\alpha_{ij}=\frac{\partial^{2}E}{\partial F_{i}\partial F_{j}}$).

The inherent asymmetry of χ(r) is very important for the partition of the molecular polarizability in terms of atomic contributions. In fact, it is clear that any subspace of $R^{3}$ cannot guarantee the symmetry of the polarizability integrated in that domain unless the domain is itself symmetric. This implies that atomic tensors may not become symmetric unless the atom sits on symmetry elements.

Despite the importance of χ(r), supramolecular chemists often attribute the interplay between molecules to an electrostatic attraction between the ground-state electron densities, hence focusing only on the hard nature of the interaction. Like for the electric potential, also for polarizability, one can simplify the information in terms of a combination of atomic contributions. A partitioning scheme is necessary to calculate atomic polarizabilities. If Ω is the atomic domain in the position space ($\Omega \subseteq R^{3}$), the components of the polarizability tensor for that domain are calculated as(6)$$\alpha_{ij}(\Omega )=\int_{\Omega }\chi_{ij}\left(r\right)d^{3}r$$

The same partition can be applied to Equation (5):(7)$$\alpha_{ij}(\Omega )=\frac{\partial \mu_{i}(\Omega )}{\partial F_{j}}$$
where $\mu_{i}(\Omega )$ is the i-th component of the atomic dipole moment μ(Ω) calculated by integrating the dipole density over the atomic domain. This consists of two parts: (a) the atomic (internal) polarization, and (b) the bond (external) polarization. The former describes the deformation of the electronic charge distribution inside the domain of atom Ω, whereas the latter is the result of charge transfer occurring between atoms, through bonds:(8)$$\mu (\Omega )=\mu_{internal}(\Omega )+\mu_{bond}(\Omega )$$

The distinction between internal and bond polarization could also be applied to the polarizability density itself, once a partition scheme is adopted. However, here, atomic tensors are calculated from the integrated atomic dipole moments.

The internal polarization is easily computed as(9)$$\mu_{internal}(\Omega )=-\int_{\Omega }\left(r-R_{\Omega }\right)\rho \left(r\right)dr$$

The bond polarization, instead, depends on the charge transferred among atoms through the bonds:(10)$$\mu_{bond}(\Omega )=\sum_{\Omega^{\prime }}\left[R_{\Omega }-R_{b}(\Omega |\Omega^{\prime })\right]Q(\Omega |\Omega \prime )$$
where $R_{b}(\Omega |\Omega^{\prime })$ is the position of the bond between atoms Ω and $\Omega^{\prime },$ and Q(Ω|Ω′) is the charge transferred through this bond, where a positive or negative charge transferred is related to atom Ω, so that $Q(\Omega |\Omega^{\prime })=-Q(\Omega \prime |\Omega )$. Another two conditions are necessary to unambiguously compute bond charges: (a) the sum of bond charges of an atom is equal to the charge of that atom; (b) in case of a loop in the molecular graph, the sum of bond charges along the loop is zero [12].

As anticipated above, the atomic tensor may not be symmetric. Otero et al. [13] attributed this feature to the way in which the atomic tensor is calculated, namely if using Equation (7) (differentiation of the integrated quantities) instead of Equation (6) (integration of the differential quantities). However, this is not the reason for the asymmetry, which depends instead on the asymmetry of the Ω domain (unless symmetry elements of the molecule intersect the domain). For visualization and exportability purposes, the symmetrization of a tensor can be easily applied by averaging $\alpha_{ij}$ and $\alpha_{ji}$ [14]. Otero et al. [13] also claimed that the procedure through Equation (7) does not take into account the regions where the polarizability density is negative, which is also not correct because they are inherently summed up in the integration of the atomic dipole moment.

The atomic polarizability is like the atomic charge q, though with the advantage of highlighting the directions along which an external electric field is more effective. In fact, in keeping with Equation (2), the atomic polarizability can be regarded as the counterpart of the atomic electric field gradient (a tensor that would describe along which directions the atomic charge is more effective). The atomic polarizabilities, therefore, may enable a simplified interpretation of secondary soft-type interactions between molecules and therefore could be used to examine and classify the vast variety of them.

In order to calculate $\mu_{interal}(\Omega )$ and $\mu_{bond}(\Omega )$, a topological graph of the system under study is necessary, because one needs to know (a) the atomic domain; (b) the charge transfer paths (bonds); and (c) the site where to locate the bond charge. The attention is therefore shifted to the partitioning, which enables both the evaluation of atomic charges and atomic polarizabilities. As well-known, a universal partition does not exist, because many schemes have been proposed in quantum chemistry to extract atoms from a molecule. Hard-space partitions offer the advantage of defining unambiguously the three loci requested above, because they identify a surface separating atoms and therefore the volume domain of the atom, the connectivity of each atom (the bonds, although not necessarily covalent bonds), and the positions of the bond charges (points on the interatomic surfaces intersecting the inter-nuclear vectors).

Among hard-space schemes, the Quantum Theory of Atoms in Molecules (QTAIM) [3] has provided one of the most physically grounded modes of partitioning, although some severe criticism has appeared in the literature [15]. The atomic domains are identified through the electron density gradient field ∇ρ. Each point in space belongs to one atomic basin Ω if the gradient field trajectory (the gradient path $r\left(s\right)$ of ρ(r) is a curve that satisfies the gradient flow equation $\frac{dr(s)}{ds}=\nabla \rho \left(r\left(s\right)\right),$ where s is a parameter along the path) $r\left(s\right)$ terminates at the attractor point of that atom, namely the electron density maximum coinciding with the position of the atomic nucleus $r_{nuc}(\Omega )$:(11)$$\left(\forall r\in \Omega \backslash \left\{r_{nuc}(\Omega )\right\}\right)\left[\underset{s\to \infty }{\lim}r\left(s\right)=r_{nuc}\right]$$

The boundaries between atoms are the interatomic (zero-flux) surfaces ∂Ω. An interatomic surface is a *locus* of points fulfilling the condition:(12)$$\left(\forall r\in \partial \Omega \right)\left[\nabla \rho \left(r\right)\cdot n(r)=0\right]$$
where n(r) are vectors normal to the surface.

The atomic basin then consists of points r satisfying the condition of Equation (11) or (12), including $r_{nuc}(\Omega ),$ and no gradient trajectory from points outside the basin Ω terminates at $r_{nuc}(\Omega )$. QTAIM has been adopted in fact for the calculation of distributed atomic polarizabilities [16] and, more recently, hyperpolarizabilities as well [17].

Other very popular schemes adopted in molecular quantum mechanics are the Hirshfeld atom (HA) [18] and the Voronoi atom (VA) [19,20]. The latter is like QTAIM, a hard-space partitioning that exactly divides the molecular space into non-overlapping atomic domains using a simple distance criterion. Each point r belongs to an atom if the nucleus is the closest nucleus to that point. Formally,(13)$$\left(\forall \;\Omega^{\prime }\neq \Omega \right)\;r\in \Omega \to d_{r-r_{nuc}(\Omega )}<d_{r-r_{nuc}(\Omega^{\prime })}$$

In Voronoi’s scheme as well, an interatomic surface ∂Ω separating atom Ω and Ω′ exists, which is the *locus* of points for which $d_{r-r_{nuc}(\Omega )}=d_{r-r_{nuc}(\Omega^{\prime })}$. The main difference between VA and the QTAIM atomic basin is that, in the former, interatomic surfaces also occur between atoms that are not chemically bonded with each other, especially if they are in the periphery of a molecule. For example, in H_2_O, an interatomic surface is shared between two H atoms, as can easily be appreciated from Figure 2. This implies that a charge transfer also occurs between these two atoms. In general, this peculiarity may significantly affect the shape of the atomic polarizabilities computed with this method. Another significant difference stands in the larger electron population computed for less electronegative (smaller) atoms. For example, H atoms in water bear 1.32 electrons for a net negative charge (−0.32), which is clearly counterintuitive. Mathematically, though, there is no ambiguity because the boundary of H atoms is positioned quite far from the nucleus (at the midpoint of any bond formed with the H atom), whereas in QTAIM, the interatomic surface is quite close to the H atoms, making their population extremely small. This feature, as well, affects the atomic polarizabilities. One additional feature of Voronoi partition is interesting: ∂Ω depends only on the molecular geometry, not on the electron density. Therefore, in the calculation of static electric polarizability (i.e., ignoring nuclear response to the field), ∂Ω is independent from the field, at variance from QTAIM. Nevertheless, the dependence of the atomic boundaries on the applied electric field can be safely neglected in the QTAIM type of atomic polarizability calculations, being extremely small.

On the other hand, the Hirshfeld atom partitioning (not to be confused with a Hirshfeld molecular surface [21]) does not rely on hard-space division. Instead, each point in space receives a contribution from every atom of the molecule. In fact, at each point r, the portion of the molecular electron density $\rho_{mol}(r)$ that belongs to atom Ω is computed with the stockholder criterion [18]:(14)$$\rho_{\Omega }(r)=\rho_{mol}(r)w_{\Omega }(r)=\rho_{mol}(r)\frac{\rho_{\Omega ,spherical}\left(r\right)}{\sum_{i=1}^{N}\rho_{i,spherical}\left(r\right)}$$
where $\rho_{i,spherical}\left(r\right)$ is the (spherical) electron density of atom i calculated from its neutral ground state in isolation; and $w_{\Omega }(r)$ is the atomic weight function, calculated as the ratio between the neutral, spherical electron density of atom Ω and the *promolecule* density, i.e., the molecular electron density calculated as the simple superposition of neutral, spherical atoms.

One disadvantage of a fuzzy partitioning like the Hirshfeld atom is that one needs to specify the bond paths in the molecule, which are not directly addressed by the partition. In general, one could consider the set of atom–atom bonds defined through the distance criterion. However, this is a pitfall of the Hirshfeld method.

**Figure 2 molecules-30-04137-f002:**
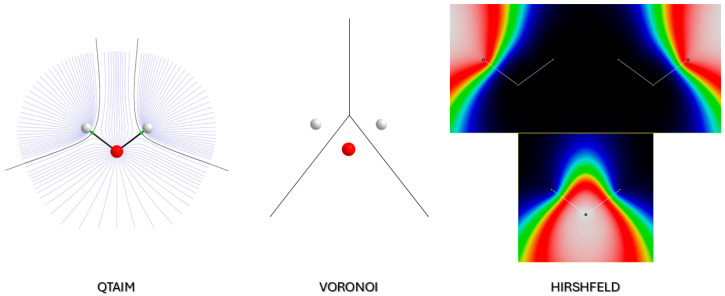
Electron density partitioning adopted in this work: (**left**) QTAIM (based on ∇ρ(r)); (**center**) Voronoi (based on distances to nuclei); (**right**) Hirshfeld (based on stockholder criterion). The molecule used for this schematic picture is H_2_O. For Hirshfeld partitioning, due to the overlapping atomic weight functions, all three $w_{\Omega }\left(r\right)$ functions are plotted separately. The color code goes from black ($w_{\Omega }\left(r\right)=0)$ to white ($w_{\Omega }\left(r\right)=1$). The gradient field map was generated with the software Aimall 19.10.12 [22], whereas the weight function bitmap was generated with MAPVIEW [23].

It is well known that charges and electric moments of atoms calculated with different electron density partitioning may differ significantly. This also has implications for their interpretations in terms of hard-type interactions, because they may not be so representative of the electrostatic potential. For example, it was demonstrated that the intermolecular electrostatic energies computed via QTAIM converge only if higher atomic multipole moments are included [24]. One may anticipate that similar discrepancies arise for the atomic polarizabilities as well. However, as discussed above, the atomic polarizabilities depend in part on the internal polarizations, which are less sensitive to the partition method. Nonetheless, in the next paragraph, the role of the partition schemes will be analyzed. Like for the atomic/molecular charges, the atomic polarizabilities sum up to the molecular polarizability. Therefore, independent from the partition scheme, it holds the following summation:(15)$$\alpha_{ij}\left(molecule\right)=\sum_{\Omega }\alpha_{ij}(\Omega )$$

Equation (15) inspires a discussion about the “exportability” of atomic polarizabilities, in other words, the similarity of atomic polarizabilities in similar chemical environments. This hypothesis enabled researchers to build databases of atomic or functional group polarizabilities [25,26] useful for the rapid prediction of molecular polarizabilities, avoiding sophisticated and time-consuming calculations. While this is valid for molecular species, so that one can also predict optical properties of molecular materials (for which the effects of crystal packing are only small perturbations), there are some interesting exceptions, for example, in ionic liquids, for which additivity [27] and linearity of the response to an external electric field [28] are lost. As a matter of fact, in ionic liquids, the ionic force is so strong that atomic polarizabilities are significantly perturbed, and therefore exportability is not possible.

## 3. Results

The methods to calculate molecular wavefunctions and atomic polarizabilities [29] are described in the Materials and Methods.

As set out in the introduction, the purpose of this work is to examine the effect of the electron density partition on the atomic polarizabilities and the possible applications in supramolecular chemistry studies, of relevance for applications in materials science.

### 3.1. Partitioning Scheme

For the first task, some prototypical molecular species, representing different intramolecular bonding types, are analyzed: simple diatomic molecules (homo- and heteropolar) and small polyatomic molecules.

The three partitioning schemes described in the background are considered:QTAIM [3]: the electron density gradient field determines the atomic domains;Voronoi [19]: atomic domains are defined based on proximity (therefore, only the molecular geometry is necessary);Hirshfeld: the stockholder [18] partition of the electron density is used.

For the application of the distributed atomic polarizability method, in the Hirshfeld partition, one must define chemical bonds (electron flow directions) and fix the position of the bond center. For the calculations reported here, bonds in the Hirshfeld method were assigned through a simple geometrical criterion (distances shorter than the sum of van der Waals radii) and bond center positioned at the interatomic midpoint, like for the Voronoi scheme. For the molecules investigated here, the QTAIM and Hirshfeld approaches identify the same network of chemical bonds. There is, however, an inherent difference in the position of the bond center, which is not the midpoint in QTAIM, but the so-called bond critical point, i.e., a point at which the electron density gradient vanishes. Only for homopolar diatomic molecules, the bond centers coincide because of the symmetry (as well as for any other bond crossed perpendicularly by at least one symmetry element of the molecule, like C-C bonds in ethane, ethene, ethine, and benzene).

#### 3.1.1. Diatomic Homopolar Molecules

Because of the inherent symmetry, the atomic polarizabilities (just like the charges and any other atomic property) are independent of the method of partition (see Table 1 and Figure 1). In fact, each component of the atomic tensors simply corresponds to(16)$$\alpha_{ij}\left(\Omega \right)=\frac{1}{2}\alpha_{ij}(molecule)$$

The molecular and atomic tensors are necessarily uniaxial, as dictated by the $D_{\infty h}$ point group symmetry and prolate due to the chemical bond, along which the component is largest. For the sake of comparison, all molecules are calculated with the bond direction oriented along the z axis. The enhancement of $\alpha_{ZZ}$ compared with $\alpha_{XX}=\alpha_{YY}$ emphasizes the role of a chemical bond for the polarizability.

Of course, in case of more bonds to an atom, one cannot easily anticipate which direction is associated with the largest polarizability, unless the molecule is linear, like $I_{3}^{-}$. Typically, the bonds with the highest covalent character feature the largest polarizability, and this is mostly responsible for the shape of the atomic tensors.

For the discussion concerning the role of halogens in supramolecular interactions, it is important to analyze, in particular, the behavior of X_2_ molecules (X = F, Cl, Br, I, see Table 1). Compared to other diatomic molecules of the same raw form in the periodic table, di-halogen molecules feature smaller bond polarizability, mostly due to the weaker chemical bond (single vs. double or triple, which occurs in other molecules like N_2_ and O_2_) and the corresponding longer distance (see, for example, the trend of N_2_, O_2_, F_2_ in Table 1).

The polarizability increases with the atomic number (thus along a column of the periodic table), in keeping with the behavior of isolated atoms [30]. For them, though, the polarizabilities are spherical, whereas in diatomic molecules, they are necessarily deformed. For all X_2_ molecules (oriented with the bond parallel to z), $\alpha_{ZZ}$ is much larger than $\alpha_{XX}=\alpha_{YY}$, but the ratio $\alpha_{ZZ}/\alpha_{XX}$ decreases along the group, likely due to the strongest covalent nature of F-F compared with interactions of the heaviest halogens. Nevertheless, for I-I, the ratio is close to 2.0, indicating that the polarizability of the species along the bond direction is much greater than the perpendicular directions. Moreover, $\alpha_{ZZ}\left(I_{2}\right)>90\;{Bohr}^{3}$. This can be considered as the *bond polarizability*, i.e., the projection of the sum of the polarizability of two bonded atoms along the bond direction [29]. It also indicates that, in principle, an iodine atom bonded to another atom features a polarizability along the direction of the bond that is larger than $45\;{Bohr}^{3}$, a value that is difficult to match by any other atom involved in covalent chemical bonds.

#### 3.1.2. Diatomic Heteropolar Molecules

In binary heteropolar diatomic molecules, the shape of the molecular polarizability tensor is like the homopolar diatomic one, because α is necessarily centrosymmetric and therefore unable to distinguish between $D_{\infty h}$ and $C_{\infty v}$ point group symmetries of the molecule. In fact, α of ΩΩ′ molecules (with Ω′≠Ω) is characterized by $\alpha_{ZZ}\neq \alpha_{XX}=\alpha_{YY},$ like for $\Omega_{2}$ molecules. However, at variance from elementary homopolar diatomic molecules, the atomic polarizabilities are not equivalent, $\alpha_{ij}(\Omega )\;\neq \;\alpha_{ij}(\Omega \prime )$. They depend on the partition scheme, like atomic charges and atomic electric moments as well.

In Table 2 and Figure 3, the results for the hypothetical molecule of LiF are presented for each of the three types of partitioning adopted. LiF is special because, at variance from all other MX molecules (M = Na, K, Rb), $\alpha_{ZZ}<\alpha_{XX}$. QTAIM assigns the largest charge separation, whereas Hirshfeld features the smallest. Consequently, with the QTAIM partition, the polarizability of the Li cation is smallest, whereas that of F is largest. Noteworthy, with Hirshfeld, the $\alpha_{ZZ}$ component of F is like that in the F_2_ molecule.

This trend is also preserved for the other molecules reported in Table 2, although the spread between various partition schemes is smaller.

**Figure 3 molecules-30-04137-f003:**
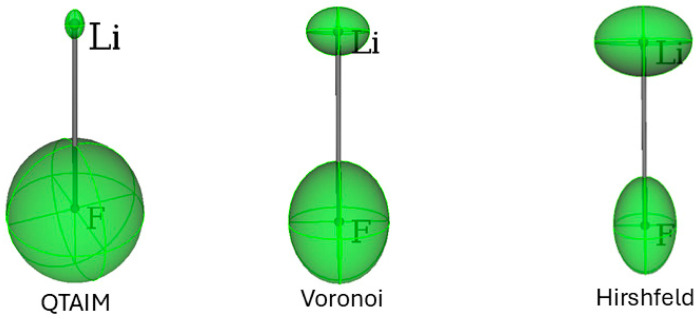
Distributed atomic polarizabilities for LiF, calculated with three different approaches to the partitioning of the same electron density distribution. Atomic ellipsoids are scaled by ${0.2\;Bohr}^{-2}$ to be representable in this plot. The plots were generated with the software Polaber 1.0 [29].

In Table 2, other heteropolar diatomic molecules are reported, including all interhalogen binary compounds of F, Cl, Br, and I. Despite evident differences among the three partition schemes for some molecules (in particular for LiF, see Figure 4), on average, the absolute discrepancy between QTAIM and HA is $1.3\;{Bohr}^{3},$ and that between QTAIM and VA is $1.2\;{Bohr}^{3}$. The standard deviations from the average of the relative differences are $1.7\;{Bohr}^{3}$ and $2.0\;{Bohr}^{3}$ (of course, the averages of relative differences are $0.0\;{Bohr}^{3}$).

In general, the atom with the largest electron population in excess of the nuclear charge is mostly deformed from sphericity by the chemical bond. This is clear because the portion of electron density that is contended between atoms (and that determines the atomic charges) is also the one more loosely bound to the nuclei, hence the most polarizable. Therefore, the atom that is assigned by the partition scheme to have the largest amount of this more polarizable electron density becomes the most polarizable atom (also depending on the number of electrons it possesses, of course).

The overall molecular polarizability itself depends on the number of electrons, and thus molecules consisting of heavier elements feature larger polarizabilities. If one normalizes the polarizabilities to the number of electrons in the molecule, one can see that $\alpha_{iso}/n_{el}$ ranges between 0.5 and 1.0 ${Bohr}^{3}/e$ (see Figure 4, top). A similar trend occurs for individual components of the molecular polarizabilities, although the range is wider (Figure 4 (bottom)). Note that, in general, atoms bearing a negative charge feature a larger $\alpha_{iso}/n_{el}$ compared to atoms with the same population but in defect with respect to the nuclear charge. While for most atoms, $\alpha_{iso}/n_{el}$ is quite independent from the partition scheme, for Li in LiF, the three different schemes return quite different values.

#### 3.1.3. Polyatomic Molecules

Here, the focus is on two simple examples, namely H_2_O and $I_{3}^{-}$. As discussed in paragraph 2, the Hirshfeld atom approach does not define the molecular connectivity; therefore, one has to introduce a bias. Although this is a pitfall, the combination of reasonable molecular graphs and HA charges and dipoles may still be useful. Of course, this issue was not relevant for diatomic molecules, given that there is only one clear bond to consider. In heteropolar diatomic molecules, the only assumption concerns the position of the bond dipole. While in hard-space partition, this is clearly located at the intersection between the interatomic vector and the interatomic surface, for fuzzy partition, the only reasonable choice is at the midpoint of the interatomic vector.

Figure 5 shows the distributed atomic polarizabilities of water molecules calculated with QTAIM, VA, and HA. Because Hirshfeld charges are much smaller than QTAIM ($q{\left(O\right)}_{HA}=-0.29$ vs. $q{\left(O\right)}_{QTAIM}=-1.17$), the polarizability of H atoms is larger (having more electrons), and consequently that of O is smaller. In Voronoi, the charges are even inverted, with H being in excess of 1 electron ($q{\left(H\right)}_{VA}=-0.37$). Consequently, polarizabilities of hydrogen atoms with VA are even larger (and that of oxygen rather small).

No other special difference is observed: in all models, the H polarizability tensor is elongated along the H-O bond (because the charge transfer overwhelms the internal polarization), whereas O polarizability is almost isotropic, taking into account that the O atom is sp^3^ hybridized and that the surrounding valence electron density is approximately tetrahedral.

$I_{3}^{-}$ is the simplest polyiodide molecule, the “forefather” of a family of molecules made by the addition of I_2_ units to $I_{n}^{-q}$ ions [31]. It is a prototype used to investigate the interplay between a hypervalent molecule and an electron donor–acceptor adduct. Being an anion, it is important to stress that the distribution of bond charges through Equation (10) is a mathematical method used to avoid the inherent origin dependence of the molecular dipole moment in a non-neutral molecule, if calculated based on charge distributions. The problem is only “fictitious” because the dipole moment of a molecule can be calculated as a first derivative of the molecular energy with respect to an applied electric field, which corresponds to calculating the molecular dipole moment using the nuclear center of charge as the origin. The method of Equation (10) inherently implies that the summation of bond dipoles returns the molecular dipole moment with respect to the center of nuclear charges. Therefore, with this method, the calculation of molecular polarizabilities with Equation (5) coincides with the double derivative of energy with respect to the field^a^ and does not depend on the chosen origin even for ionic molecules.

As one can see from Figure 6, $I_{3}^{-}$ features a huge polarizability along the molecular axis z, so that $\alpha_{ZZ}=307\;{Bohr}^{3}$ is much larger than $\alpha_{XX}=\alpha_{YY}=87\;{Bohr}^{3}$. This is also reflected by the atomic polarizabilities. In this test case, the three partition schemes return a very similar picture, with only minor differences. The representation of $Tr\left[\chi (r)\right]$ clearly addresses the regions of higher polarizability at the two external atoms.

### 3.2. Coordination Complexes

The electron donor–acceptor bond strongly depends on the polarizability of the species involved. The donating group is electron rich and exerts an electric force on the acceptor group, which polarizes in such a way to avoid electron repulsion with the donor and favor an electrostatic attraction. The bond strength inherently depends on this ability of the acceptor group to polarize. The mechanism is particularly relevant in metal complexes, especially because together with the ligand-to-metal donation, a retro-donation from the metal may occur as well.

Figure 7 shows calculations for a prototype of organometallic molecules, namely chromium hexacarbonyl, Cr(CO)_6_, first reported by Job and Cassal [32]. On top, the polarizability density of the complex is shown together with those of the two fragments, Cr(CO)_5_ and CO, calculated in isolation (i.e., without considering the other fragment). The two species feature a broad region of positive $Tr\left[\chi (r)\right]$, located on the C atom of CO and Cr atom of Cr(CO)_5_. This feature is important for both ligand-to-metal donation and metal-to-ligand back-donation. Upon complexation, the region of the C-Cr bond remains highly polarizable, but the C-O bond now features a region of negative polarizability, i.e., a region that would react oppositely to an external electric field. Negative polarizability density regions are already present in all C-O bonds of the Cr(CO)_5_ fragment, but not visible in the isolated CO, at least with the selected isosurface level (−0.075). In Figure 1, instead, the region is visible, because it is plotted at a lower isosurface level (−0.03), which indicates that the complexation to Cr has enhanced this feature.

In terms of the integrated atomic polarizability tensors (Figure 7 (bottom)), one can see that CO in isolation is characterized by a strongly bond-prolated O ellipsoid and a more isotropic C polarizability. On the other hand, the Cr(CO)_5_ fragment features an oblate ellipsoid for Cr, due to the missing coordination along one direction. Upon complexation, the Cr-C bonds (all necessarily identical by symmetry) display strong polarizability of the C atom along the bond. This direction coincides with the C-O bond, but, as discussed above, $Tr\left[\chi (r)\right]$ is slightly negative in some region of this bond.

To explain this result, one should remember that the atomic polarizability tensor comes from the integration over space of  χ(r), which is the sum of a strongly positive polarizability of the Cr-C bond and an alternating negative and positive polarizability along the C-O bond. For this reason, a combined representation of $Tr\left[\chi (r)\right]$ and distributed atomic polarizability tensors is preferable because it provides a more comprehensive view of the system.

In Figure 8, a similar analysis is presented for the iodide of iodammonium (NH_3_I_2_), as originally named by Guthrie [33]. Here, the donation occurs only in one direction, from ammonia to iodine. It is noteworthy that NH_3_ even in isolation has a negative polarizability density on the side of the lone pair at the nitrogen. However, this is not relevant for the eventual formation of a complex with iodine, because it is the electron acceptor group (I_2_) that must be prompted to repolarize upon complexation.

### 3.3. Secondary Interactions

In a previous article [34], the distributed atomic polarizabilities were adopted for the analysis of hydrogen bonds, and the study investigated the implication in materials properties. Here, I focus instead on the possibility to use the polarizability density and distributed atomic polarizabilities for the interpretation of supramolecular interactions from the soft-side perspective.

In Figure 7, a new kind of visualization is proposed for the polarizability density, like the very common representation of the electrostatic potential of a molecule on an isosurface of the electron density. Instead of $\phi \left(r\right)$, $Tr\left[\chi (r)\right]$ is mapped on the electron density isosurface (typically $\rho \left(r\right)=0.1\;{eBohr}^{-3}$ and $Tr\left[\chi (r)\right]$ ranging from −0.05 to 0.3). The four molecules selected for the visualization address interesting features. First, one can easily appreciate that the classical σ- and π-holes appearing in electrostatic potential maps [35] are also addressed by $Tr\left[\chi (r)\right]$. For example, in C_2_H_4_, the blue regions (large, positive polarizability density) are close to the carbon nucleus and perpendicular to the C=C bond, which are analogous to the π-holes visible through the electrostatic potential [36]. In ClF and SeF_2_, instead, a feature appears similar to the σ-hole typical of strong σ-bonds and responsible for the directional electrophilicity of those atoms. It is the same for the H atoms in H_2_O. These features should not be interpreted from the point of view of a hard-type interaction (electrostatic attraction with electron-rich atoms of a vicinal molecule), but from the point of view of a soft-type interaction with polarizable electron pairs of other molecules. Interestingly, the two different machineries share the same requirements for the incoming nucleophilic molecule: it must have a region of strong charge concentration. The similar shape and location of the highest polarizability density and the σ-holes is quite clear if one considers that the electron density depletion, causing the positive electrostatic potential opposed to a σ bond, is due to the electron density polarization along that bond, which is also the same direction of an electron density polarization due to the interplay with the antibonding $\sigma^{*}$ orbital. Similar reasoning holds for the π-holes. Therefore, the polarizability density features mapped in Figure 9 are better understood as $\sigma^{*}$ and $\pi^{*}$-holes.

One drawback of visualizing the polarizability density on the electron density isosurface is that one cannot fully appreciate the directionality of the polarizable density. Therefore, a picture of the distributed atomic polarizabilities may be associated to complement the information. For example, the plot of SeF_2_ in Figure 9 emphasizes the fact that F atoms are more polarizable in their valence shell, especially opposed to Se-F bonds, whereas Se is not much polarizable in the valence region. Its larger atomic polarizability emerging from Figure 10 comes from the larger number of electrons of this atom. On the other hand, Figure 10 more straightforwardly indicates that F is polarizable along the Se-F bond, whereas Se is not so affected by the chemical bonds, and its polarizability is almost isotropic. In Figure 10, one can see that similar features occur for AsF_3_ and GeF_4_.

## 4. Discussion and Conclusions

While the fundamental quantity for the analysis of the induced polarization of a molecular electron density is the polarizability density χ(r) (a tensor function), the results presented in the previous section indicate that other more easily interpretable quantities could be investigated. For example, $Tr\left[\chi (r)\right]$ is a scalar 3D function, representing the isotropic polarizability density ($Tr\left[\chi (r)\right]/3$), and therefore it addresses regions of space that are more polarizable. It does not indicate along which direction the electron density will preferentially polarize upon application of an electric field, information which is instead provided by the distributed atomic polarizabilities α(Ω), which are atom-centered tensors coming from the integration $\chi \left(r\right)$ within atomic domains. On the other hand, α(Ω)s are limited because they convey information about the overall polarizability of an atom, without addressing specifically if the valence region is polarizable, which is fundamental information for the prediction of supramolecular interactions. This is the reason why a combination of $Tr\left[\chi (r)\right]$ and α(Ω) is more effective for the interpretation of the molecular susceptibility to interact or react. An analogy can be drawn between polarizability density + polarizabilities and electrostatic potential + atomic charges (the two most diffused indicators to study the hard-type electrostatic intermolecular interactions). Atomic charges give an idea about the possible reactive atom in a molecule, while electrostatic potential more precisely addresses the region of space that imparts an electrophilic or nucleophilic character. The analogy is however not completely fair, because the atomic polarizabilities are tensors, and hence they inherently contain information about the directionality of a possible interaction with vicinal molecules, whereas atomic charges are simply scalars. Indeed, the best analogy for the atomic polarizabilities would be with the atomic electric field gradient $\nabla_{r}F$, a quantity that is seldom used and anyway only computed/measured at nuclear sites (instead of being integrated over an atomic domain).

In this work, I have tested different partition schemes to calculate α(Ω), given that there is no unique definition of an atom in a molecule. Two hard-space partitions (QTAIM and Voronoi) were used as well as one fuzzy scheme, consisting of overlapping densities like Hirshfeld.

The partition mainly influences atomic-derived charges, and hence the bond dipoles, which depend on charge transfer between pairs of atoms. For this reason, in the case of strongly ionic compounds, like LiF, the atomic polarizabilities may significantly differ because the atomic charges derived with different schemes are in fact extremely divergent. Figure 3 and Figure 4 (bottom) clearly illustrate these discrepancies. Of course, the unavoidable variance of atomic populations produced by different definitions of an atom in a molecule affects the distributed polarizabilities as well. Nevertheless, some considerations about the suitability of the different methods are useful. The evaluation of atomic polarizabilities requires not only the definition of an atom in a molecule but also of the “channels” for electron density shifts and the precise point at which to fix a bond dipole. In this respect, QTAIM is seamlessly ideal because the topological analysis of the electron density returns atomic domains, interatomic bond paths and surfaces, and, especially, the bond critical points, where the bond dipole is positioned. Voronoi’s partition also offers atomic domains and interatomic surfaces, but one has to assume that the dipole moment is located at the intersection of the internuclear vector and the interatomic surface. Another drawback of the VA scheme is that interatomic surfaces also occur between atoms that are not chemically bonded (like the two H atoms in H_2_O). This inherently opens up channels that are not physically sensible. Additionally, the VA scheme tends to assign too many electrons to less electronegative atoms, which implies larger atomic polarizabilities for those atoms.

Compared to QTAIM, VA is computationally less expensive, because the determination of the atomic domain is quite rapid, at variance from QTAIM.

If one instead considers the Hirshfeld atom partition, more biases arise because no interatomic boundary can be defined, and therefore one has to assume chemical bonds based on a prejudice, and position the bond dipole at the midpoints of the internuclear vectors. Nevertheless, the Hirshfeld approach has some advantages. It is very easily computed because one does not need to define atomic spatial domains. Moreover, the results are not enormously different from QTAIM, unless for those systems for which the partition significantly affects the atomic charge (see, for example, LiF). Of course, instead of the simple HA, one could use the iterative Hirshfeld partition [37] that typically assigns atomic charges more consistent with chemical expectation. These alternative schemes can be easily implemented as well as any other scheme, considering that the mathematics presented in Section 2 are valid independently from the type of partitioning.

Some comments are also necessary for the proper interpretation of the atomic polarizabilities. As explained in Section 2, any (hard-space or fuzzy) partition Ω of the $R^{3}$ space may produce a subspace that is not as symmetric as an ellipsoid (or, in general, a symmetric second-order tensor). To be sufficiently symmetric, the point group symmetry of the Ω subspace should be at least $C_{2v}$. Because the weight function in the case of HA is defined in the entire $R^{3}$, the condition to obtain a symmetric tensor is that w(r) be at least of $C_{2v}$ point group symmetry, which again means that the atom sits on a special symmetry position in the molecule (which is typically an exception, not the rule). If the atomic domain has lower symmetry, the calculated tensor is not symmetric, which produces some disadvantages. The tensor symmetrization of Nye [14] solves the problem because it does not alter the major directions of polarizability of an atom, and it also guarantees that the overall summation of atomic polarizabilities coincides with the molecular polarizability. In fact, the symmetrization procedure creates an antisymmetric tensor as well, but it was demonstrated [29] that the sum of all antisymmetric atomic tensors in a molecule vanishes, and hence the sum of symmetrized tensors must coincide with the molecular tensor. The a posteriori symmetrization enables an easy visualization and therefore interpretation of the atomic contribution, and it also allows the exportability of polarizabilities. In fact, databases of so-defined atomic polarizabilities have recently appeared [25,38], demonstrating the possible applications in crystal engineering.

In this work, I have not examined the level of theory (type of DFT functionals, ansatz of the wavefunction, etc.) or the basis set used for the quantum chemical calculations of molecular electron densities and their field derivatives. This is, of course, a very important and critical step, but it is out of the scope of this paper, which is analyzing the effect of the atom partitioning scheme. Of course, the greater the accuracy of the molecular wavefunction (and hence the electron density), and, especially, the wavefunction calculated under an electric field, the greater the accuracy of the calculation of the polarizability density and the distributed atomic polarizabilities. The reader can find more information about the effect on atomic and molecular polarizabilities due to the level of theory and the basis set in a previous work, focusing on amino acids and their optical properties [39]. The general conclusions of this article, though, are not affected by the type of quantum chemical calculation, because they concern only the methodology to divide the polarizability density and its more appropriate application. A finer comparison between post-Hartree–Fock calculations and DFT calculations or between different types of basis sets will be discussed in future work.

In conclusion, this work has demonstrated the advantages of using distributed atomic polarizabilities and polarizability density functions to interpret molecular features that are fundamental in molecular recognition processes and reactivities. The focus on electric polarizability instead of electric potential or charges shifts the perspective from hard to soft interactions.

The tools described in the previous sections are highly applicable for analyzing intermolecular interactions, which are becoming increasingly sophisticated, especially given the emerging applications in materials science. Therefore, the analysis based on properties directly correlated with soft interactions is extremely important and should not be underrated.

## 5. Materials and Methods

### 5.1. Molecular Wavefunction Calculations

All molecular geometries were optimized using the hybrid exchange–correlation functional CAM-B3LYP [10] featuring long-range correction for the exchange potential. This functional was selected because, as demonstrated in previous work [39], for polarizabilities, it has excellent agreement with post Hartree–Fock methods. For all calculations, the basis set adopted was quadruple-zeta valence including the polarization function (QZVP) [40]. For all molecules, the geometries were optimized, and then molecular polarizabilities were calculated using coupled, perturbed Kohn–Sham calculations. For the distributed atomic polarizabilities, the coupled, perturbed Kohn–Sham molecular wavefunctions were calculated under an electric field of 0.005 atomic units (a.u.) applied along each coordinate ±x, ±y, ±z.

All calculations were carried out with Gaussian16 [41].

### 5.2. Electron Density Partitioning and Distributed Polarizabilities

The QTAIM partitioning was calculated with the software AimAll [22]. The Hirshfeld atom partitioning was calculated with Gaussian16 [41]. The Voronoi partitioning was calculated with a locally written routine *poladens*, based on numerical integration. The polarizability density was also calculated by numerical differentiation with *poladens*. The routine is available from https://github.com/pieromacchi/Poladens.git (accessed on 30 September 2025).

The atomic polarizabilities were calculated with the software Polaber [29].

## Figures and Tables

**Figure 1 molecules-30-04137-f001:**
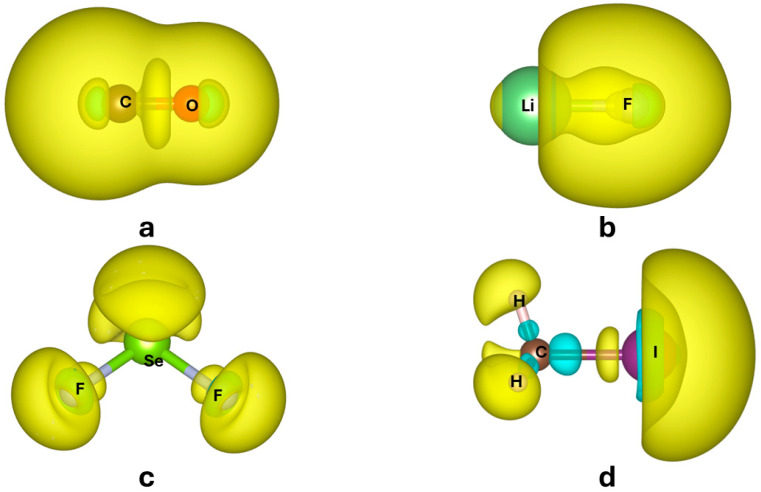
$Tr\left[\chi (r)\right]$ of CO (**a**) LiF (**b**), SeF2 (**c**), and CH3I (**d**) calculated with CAM-B3LYP [10] density functional and QZVP basis set. Isosurfaces corresponding to positive values of χ(r) are in light yellow, negative in cyan. Values of the isosurfaces change from molecule to molecule for a better visualization of the shape of the polarizability density (namely, ±0.03 for CO, ±0.02 for LiF, ±0.1 for SeF2, and ±0.09 for CH3I). Positive surfaces in yellow, negative surfaces in cyan. The software VESTA3.5.7 [11] was used to produce these plots.

**Figure 4 molecules-30-04137-f004:**
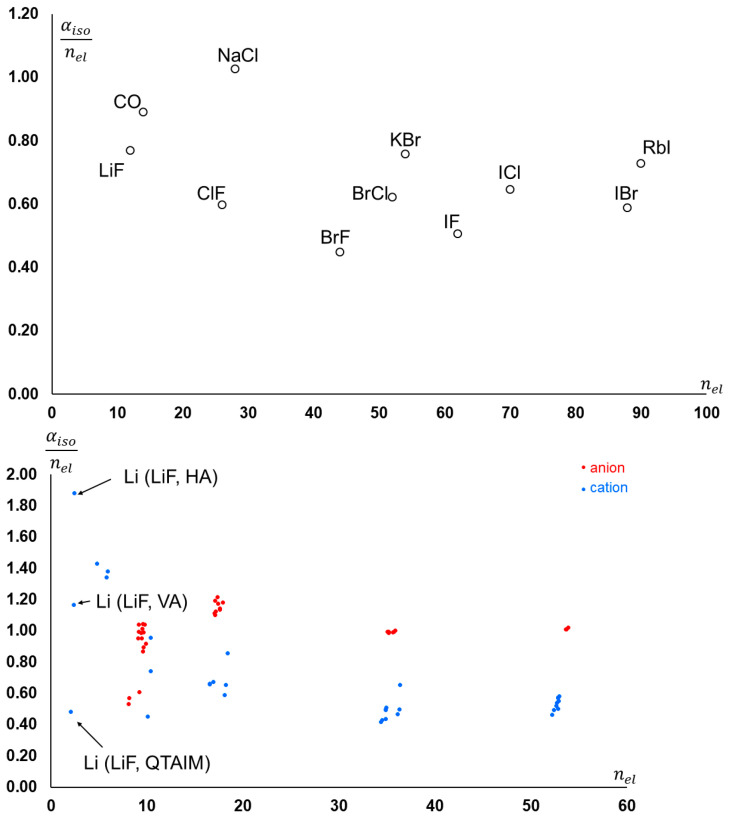
(**Top**): Isotropic molecular polarizability per number of electrons for the molecules reported in Table 2. (**Bottom**): Atomic isotropic polarizability per number of electrons (atomic population) in the same molecules. Anions and cations are represented in different colors.

**Figure 5 molecules-30-04137-f005:**
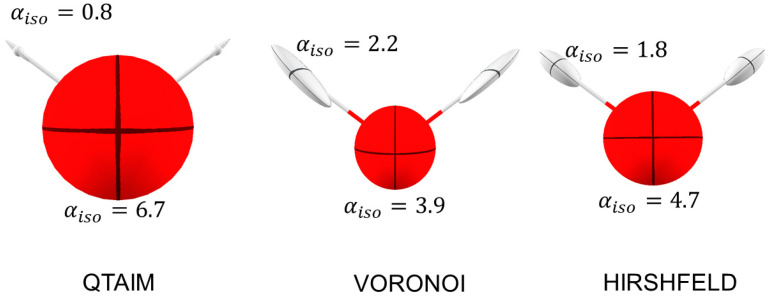
The distributed atomic polarizabilities in H_2_O molecule calculated with various partition schemes. Isotropic atomic polarizabilities are indicated in ${Bohr}^{3}$. The ellipsoids are scaled by ${0.3\;Bohr}^{-2}$ to be representable in this plot. Note that with QTAIM partition, the H ellipsoid perpendicular to the bond is even smaller than the size of the stick used to visualize the bond. The plots were generated with the software Polaber [29].

**Figure 6 molecules-30-04137-f006:**
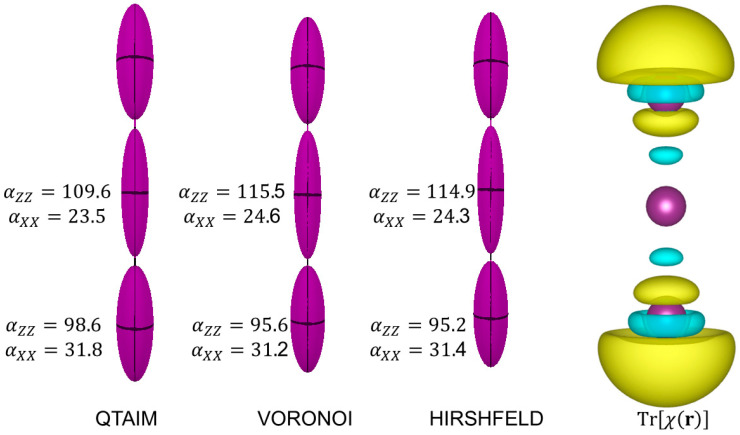
The distributed atomic polarizabilities in $I_{3}^{-}$ calculated with various partition schemes. Isotropic atomic polarizabilities are indicated (in ${Bohr}^{3}$). The ellipsoids are scaled by ${0.05\;Bohr}^{-2}$ to be representable in this plot. The plot on the right is the polarizability density $Tr\left[\chi (r)\right]$, represented with isosurfaces of ±0.2 (dimensionless) with positive surfaces in yellow and negative surfaces in cyan. The distributed polarizabilities were created with the software Polaber [29], whereas $Tr\left[\chi (r)\right]$ isosurfaces are generated with VESTA3 [11].

**Figure 7 molecules-30-04137-f007:**
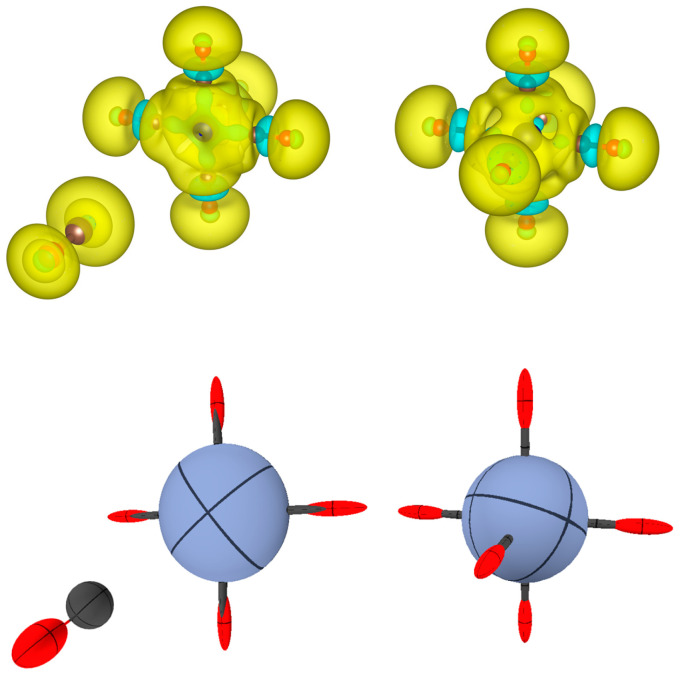
(**Top**): $Tr\left[\chi (r)\right]$ for Cr(CO)_6_ and for the fragments CO and Cr(CO)_5_. $Tr\left[\chi (r)\right]$ is represented with isosurfaces of ±0.075 (dimensionless), with positive surfaces in yellow and negative surfaces in cyan. (**Bottom**): The same representation in terms of distributed atomic polarizabilities. All atomic tensors are scaled by $0.2\;{Bohr}^{-2}$ for sake of visualization. Note that the C atoms coordinated to Cr feature a strongly prolated tensor (that looks like a bond stick in the picture).

**Figure 8 molecules-30-04137-f008:**
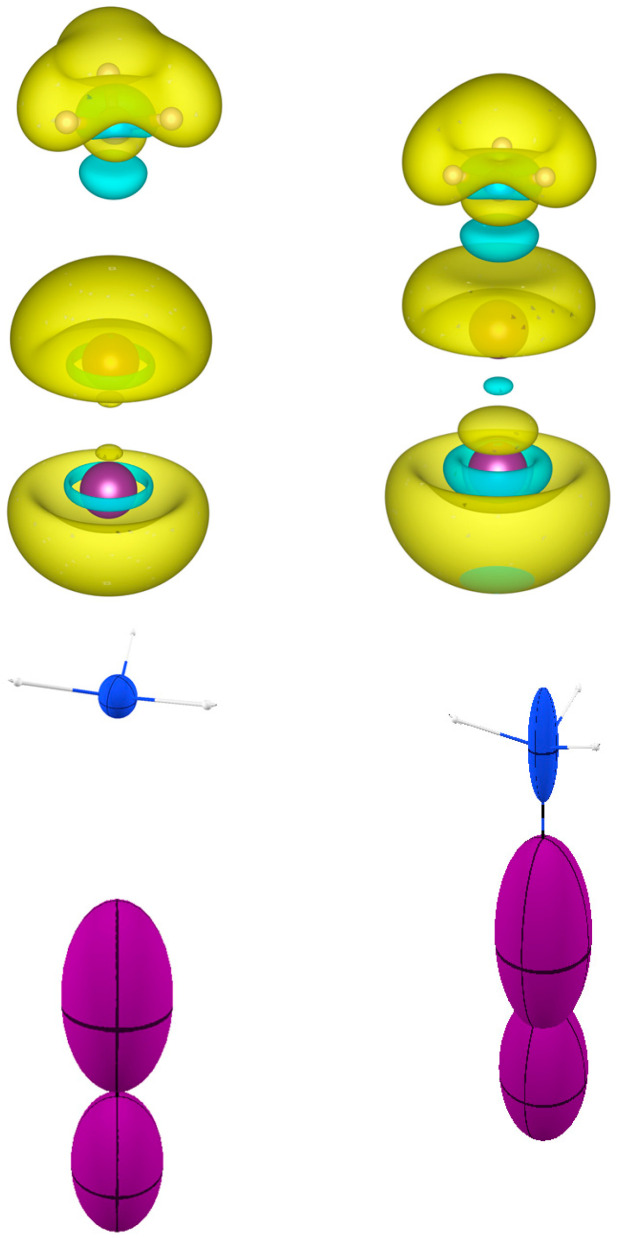
(**Top**): $Tr\left[\chi (r)\right]$ for NH_3_I_2_ and for the isolated fragments NH_3_ and I_2_. $Tr\left[\chi (r)\right]$ is represented with isosurfaces of ±0.1 (dimensionless), with positive surfaces in yellow and negative surfaces in cyan. (**Bottom**): The same representation in terms of distributed atomic polarizabilities. All atomic tensors are scaled by $0.1\;{Bohr}^{-2}$ for the sake of visualization.

**Figure 9 molecules-30-04137-f009:**
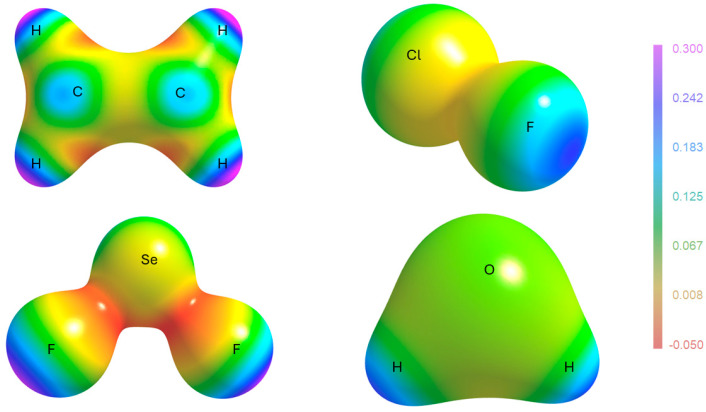
The polarizability density plotted on the electron density isosurface ($\rho \left(r\right)=0.1\;e{Bohr}^{-3}$) for C_2_H_4_, ClF, SeF_2_, and H_2_O. The color-coded scale of $Tr\left[\chi (r)\right]$ is the same for all molecules and given on the right. It is notable that all σ bonds are associated with a large polarizability density opposed to the bonds (in keeping with the analogous feature of the electrostatic potential). For the C=C double bonds instead, the largest polarizability is perpendicular to the bond (in keeping with the famous π-hole of the electrostatic potential). In C_2_H_4_ and SeF_2_, regions of negative polarizability density are visible, along C-H and Se-F bonds. In these regions, the reaction of the electron density towards an electric field is reversed.

**Figure 10 molecules-30-04137-f010:**
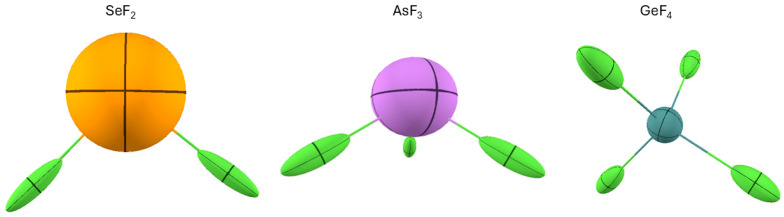
The distributed atomic polarizabilities in three fluoride molecules: SeF_2_, AsF_3_, and GeF_4_. All atomic polarizabilities are scaled by $0.2\;{Bohr}^{-2}$ to be visualized in the plot. The visuals were created with the software Polaber [29].

**Table 1 molecules-30-04137-t001:** Molecular and atomic polarizabilities (in Bohr^3^) in homopolar diatomic molecules calculated at CAM-B3LYP/QZVP level. For these molecules, the atomic polarizabilities are independent from the partitioning scheme.

Molecule	$\alpha_{XX},\alpha_{YY}$	$\alpha_{ZZ}$	$\alpha_{XX}\left(A\right),\alpha_{YY}(A)$	$\alpha_{ZZ}(A)$	$\alpha_{XX}\left(B\right),\alpha_{YY}(B)$	$\alpha_{ZZ}(B)$
N_2_	8.9	14.8	4.4	7.4	4.4	7.4
O_2_	6.1	13.2	3.0	6.6	3.0	6.6
F_2_	4.3	11.4	2.1	5.7	2.1	5.7
Cl_2_	20.6	39.5	10.3	19.8	10.3	19.8
Br_2_	28.2	57.5	14.2	28.8	14.2	28.8
I_2_	51.6	94.8	21.8	47.4	21.8	47.4

**Table 2 molecules-30-04137-t002:** Molecular and atomic polarizabilities (in Bohr^3^) in heteropolar diatomic molecule AB calculated at CAM-B3LYP/QZVP level. Three partition methods are compared: QTAIM, Voronoi (V), and Hirshfeld atom (HA). Q is the charge of the atoms (positive atom is the first in the formula).

Molecule A-B	$\alpha_{XX},\alpha_{YY}$	$\alpha_{ZZ}$		Q	$\alpha_{XX}\left(A\right),\alpha_{YY}(A)$	$\alpha_{ZZ}(A)$	$\alpha_{XX}\left(B\right),\alpha_{YY}(B)$	$\alpha_{ZZ}(B)$
LiF	9.4	8.9	QTAIM	±0.92	0.8	1.4	8.7	7.5
			V	±0.63	3.0	2.3	6.4	6.6
			HA	±0.57	5.3	3.1	4.2	5.8

NaCl	25.8	34.6	QTAIM	±0.89	3.2	7.3	22.7	27.6
			V	±0.63	6.8	9.5	18.4	25.3
			HA	±0.58	9.9	10.1	15.9	24.8

KBr	35.6	51.5	QTAIM	±0.89	8.1	15.9	27.6	36.2
			V	±0.75	9.7	16.5	24.5	35.3
			HA	±0.62	14.9	17.5	20.8	34.5

RbI	56.9	82.6	QTAIM	±0.89	11.9	26.8	45.1	57.4
			V	±0.72	13.5	27.3	40.0	55.5
			HA	±0.63	21.3	28.8	35.9	55.3

ClF	12.9	20.8	QTAIM	±0.43	10.6	11.8	2.3	9.1
			V	±0.46	10.0	12.7	2.9	8.1
			HA	±0.10	10.6	13.0	2.3	7.8

BrF	16.8	25.6	QTAIM	±0.51	14.4	15.7	2.4	9.9
			V	±0.43	13.5	16.2	3.3	9.4
			HA	±0.15	14.6	16.6	2.3	9.0

IF	29.5	35.2	QTAIM	±0.62	26.8	24.3	2.7	10.9
			V	±0.80	24.6	24.1	4.8	11.1
			HA	±0.18	27.2	24.9	2.4	10.3

BrCl	24.5	47.8	QTAIM	±0.15	14.0	24.2	10.5	23.6
			V	±0.13	13.8	24.9	10.7	22.9
			HA	±0.05	14.1	25.0	10.4	22.8

ICl	36.7	62.2	QTAIM	±0.33	25.7	33.8	11.0	28.5
			V	±0.39	24.9	34.9	11.6	27.3
			HA	±0.09	26.2	35.2	10.5	27.0

IBr	40.4	74.6	QTAIM	±0.20	25.9	38.9	14.6	33.8
			V	±0.31	25.3	39.9	14.9	34.6
			HA	±0.04	26.1	40.2	14.3	34.4

CO	11.2	15.0	QTAIM	±1.20	6.8	7.0	4.4	8.0
			V	±0.19	7.6	8.2	3.6	6.8
			HA	±0.08	8.0	8.5	3.2	6.5

## Data Availability

No new data were created or analyzed in this study. Data sharing is not applicable to this article.

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
