# Peer review of "Polarizabilities of Atoms in Molecules: Choice of the Partitioning Scheme and Applications for Secondary Interactions"

_molecules, 2025, doi:10.3390/molecules30204137_

Round 1

Reviewer 1 Report

Comments and Suggestions for Authors

In this study, the authors analyzed in detail the advantages of using distributed atomic polarizabilities and polarizability density functions for the interpretation of molecular features that could be fundamental in molecular recognition processes and reactivities. The tools described in this study may be useful in the analysis of intermolecular interactions. Although the theoretical foundation of this study has significant issues, after the authors make the following modifications, I believe the article is still worthy of publication.

  1. On lines 52 and 53 of the article, as well as in several other places, the author claims that 'As introduced above, the most important quantities for the evaluation of the intermolecular interactions are the electric potential and the polarizability density'. Such a claim is overly dogmatic. It is well known that for uncharged and nonpolar species, the long-range interaction is dominated by dispersion energy. The author should specifically explain in the article why the contribution of dispersion is neglected.
  2. Section '3.2. Secondary interactions' is overly simplistic, and readers actually gain little valuable information from it. The discussion in this section should be strengthened.
  3. There are numerous grammatical errors in the article, and the author should carefully revise them. Here are just two examples. On line 616 of the article, 'analyzed in details' should be changed to 'analyzed in detail' ('detail' does not require plural form when used as an uncountable noun); on line 618, 'that could result fundamental' has a collocation issue, and it is suggested to be modified to 'that could be fundamental' or 'that could prove fundamental' ('result' is rarely directly linked with adjectives when used as a linking verb).

Author Response

Comment 1: On lines 52 and 53 of the article, as well as in several other places, the author claims that 'As introduced above, the most important quantities for the evaluation of the intermolecular interactions are the electric potential and the polarizability density'. Such a claim is overly dogmatic. It is well known that for uncharged and nonpolar species, the long-range interaction is dominated by dispersion energy. The author should specifically explain in the article why the contribution of dispersion is neglected.

Rresponse 1: dispersion energy, according with the famous London theory, depends in fact on the polarizability of both interacting molecules . (IA, IB, are ionization potentials, R is the intermolecular distance).
Therefore, this study does not neglect at all the important contribution of dispersion energy and the investigation of atomic polarizabilities is in fact a way to appreciate better which sites are more subject not only to induced polarization but also to London forces. Anyway, to stress this, I have now included a sentence in the introduction.

Comment 2:  Section '3.2. Secondary interactions' is overly simplistic, and readers actually gain little valuable information from it. The discussion in this section should be strengthened.

Response 2: In section 3.2 a new view of the classical concepts of sigma and pi-holes is proposed. I do not understand why the referee considers this over simplistic, given that it has unprecedented ideas that have never been proposed before (sigma and pi-holes being interpreted only in terms of electrostatic potential). Anyway, the novelty of this approach has been better highlighted. Moreover, in view of comments of referee 3, this section has now become 3.3

Comment 3:  There are numerous grammatical errors in the article, and the author should carefully revise them. Here are just two examples. On line 616 of the article, 'analyzed in details' should be changed to 'analyzed in detail' ('detail' does not require plural form when used as an uncountable noun); on line 618, 'that could result fundamental' has a collocation issue, and it is suggested to be modified to 'that could be fundamental' or 'that could prove fundamental' ('result' is rarely directly linked with adjectives when used as a linking verb).

Response 3: I have carefully checked grammar errors and corrected the text. I note that all the other 4 referees did not address any issue with the language. Refuses are of course accidents and were corrected to the best of my knowledge.

Reviewer 2 Report

Comments and Suggestions for Authors

This manuscript presents a thorough and well-structured discussion of how molecular polarizabilities can be partitioned into atomic contributions, with a particular focus on the comparison between different partitioning schemes (QTAIM, Voronoi, Hirshfeld) and the use of the polarizability density as a tool to analyze secondary interactions. The study is conceptually clear, methodologically rigorous, and the numerous examples provided (from simple diatomic molecules to polyatomic cases and supramolecular interactions) convincingly illustrate the proposed framework.

While the manuscript is already of high quality, I recommend the author to broaden the scope of the discussion by explicitly connecting it to recent advances in the field of ionic liquids, where charge delocalization and non-additivity in polarizabilities have been shown to play a crucial role. Three references are particularly relevant and should be cited and discussed:

(1) J. Mol. Liq. 2021, 117099: The finding that polarizabilities in ionic liquids are strongly non-additive due to electronic delocalization complements the present manuscript’s argument that partitioning is non-trivial and context-dependent.

(2) J. Mol. Liq. 2022, 349, 118153: This work highlights how delocalization effects influence nonlinear response properties in ionic liquids, which ties directly into the present article’s emphasis on polarizability density as a descriptor of soft interactions.

By integrating these references, the author would not only strengthen the connection between fundamental theoretical developments and applied contexts (such as ionic liquids and functional materials), but also situate the present work within a broader ongoing discussion about charge delocalization, non-additivity, and environment-dependent response properties.

Overall, I find this manuscript to be an excellent contribution to the field. With the inclusion of the above literature and a short discussion linking the present methodology to charge delocalization effects in ionic liquids, the work will be even more compelling to a broad readership. I recommend publication after these minor revisions.

Author Response

I thank this reviewer for the report. The main comment concerns the literature and in particular adding some reference

Comment 1:  While the manuscript is already of high quality, I recommend the author to broaden the scope of the discussion by explicitly connecting it to recent advances in the field of ionic liquids, where charge delocalization and non-additivity in polarizabilities have been shown to play a crucial role. Three references are particularly relevant and should be cited and discussed:

(1) J. Mol. Liq. 2021, 117099: The finding that polarizabilities in ionic liquids are strongly non-additive due to electronic delocalization complements the present manuscript’s argument that partitioning is non-trivial and context-dependent.

(2) J. Mol. Liq. 2022, 349, 118153: This work highlights how delocalization effects influence nonlinear response properties in ionic liquids, which ties directly into the present article’s emphasis on polarizability density as a descriptor of soft interactions.

Response 1: The two references were added, together with a small discussion on additivity and exportability of the polarizabilities at the end of section 2. The referee mentioned three references to add, but addressed only two articles, that anyway I think are quite representative for ionic liquids.

Reviewer 3 Report

Comments and Suggestions for Authors

The manuscript “Polarizabilities of atoms in molecules: choice of the partitioning scheme and applications for secondary interactions” explores how molecular polarizability can be partitioned into atomic contributions to better understand chemical reactivity and intermolecular interactions. It compares three partitioning schemes—QTAIM (Quantum Theory of Atoms in Molecules), Voronoi, and Hirshfeld—and evaluates their impact on atomic polarizability tensors. The author introduces the concept of polarizability density, a tensor function that maps how electron density responds to external electric fields, offering a complementary perspective to electrostatic potential in predicting soft-type interactions.

Through computational analysis of diatomic and polyatomic molecules, the study demonstrates that atomic polarizabilities vary significantly depending on the partitioning method, especially in ionic systems like LiF. QTAIM is highlighted for its physical rigour and ability to define atomic domains and bond paths, while Voronoi and Hirshfeld offer computational simplicity but introduce interpretive biases.

The manuscript also proposes using polarizability density maps and distributed atomic polarizabilities to visualise regions of high reactivity, such as σ- and π-holes, which are crucial for supramolecular recognition. These tools are positioned as valuable for crystal engineering, materials design, and understanding secondary interactions. The work concludes that combining polarizability density with atomic tensors provides a robust framework for analysing molecular susceptibility to polarisation and interaction.

I think the current manuscript could be improved with the following comments:

  • Readers of the manuscript may find themselves somewhat confused as ambiguous or undefined terms (e.g., "polarizability density" and "soft-type interactions") are occasionally used throughout the manuscript without a sufficient theoretical basis. Clearly defining these concepts at the beginning of the text and using them consistently throughout would improve understanding.
  • Some sections, particularly those dealing with QTAIM and Hirshfeld schemes, assume familiarity with advanced partitioning methods. As with the previous comment, including a brief comparative summary or outline would facilitate accessibility for a wider audience.
  • While, in view of the abstract and the title of the manuscript, a reader might expect to find a discussion of the applications of secondary interactions, particularly for electron donor-acceptor complexes, this is not the case; the content provided focuses mainly on a theoretical comparison of partitioning schemes (QTAIM, Voronoi, and Hirshfeld) and their application to simple homopolar and heteropolar diatomic molecules. This poses some challenges in understanding the manuscript, since the crucial "applications" section is either missing or incomplete; in particular, the use of polarizability density as a predictive tool for secondary interactions is promising but underdeveloped. A more explicit comparison with electrostatic potential-based methods would clarify its added value.
  • While the comparison of partitioning schemes is well-executed, the manuscript could benefit from a more in-depth discussion on the practical implications of the observed differences. The paper correctly notes that the atomic polarizabilities depend on the partitioning scheme. But do not specify whether frequency-dependent polarizabilities were considered or whether static values were used exclusively. This distinction is critical for interpreting secondary interaction potentials. The discussion could be enhanced by providing a clear, expert-level perspective on how a computational chemist would choose one method over another for a specific supramolecular system and what the physical meaning of the resulting differences truly is.
  • Figure 1 displays the trace of polarizability density, providing a visualisation of the data. Likewise, Figure 3 offers a visual comparison of atomic polarizability ellipsoids for LiF using different schemes. The manuscript could incorporate similar visual representations for secondary interaction applications to illustrate the utility of the distributed atomic polarizability concept. Additionally, the resolution of certain polarizability maps may not be sufficient to distinguish subtle features, especially in polyatomic systems. Higher-quality renderings or zoomed-in panels would be useful. Clear legends and consistent formatting across all figures are also recommended.

The manuscript presents an intriguing and potentially impactful approach to analysing molecular polarizability through atom-based partitioning schemes. However, to meet the scientific and editorial standards, some revisions are required.

Author Response

I thank this referee for the detailed comments. Please see below the responses.

Comment 1: Readers of the manuscript may find themselves somewhat confused as ambiguous or undefined terms (e.g., "polarizability density" and "soft-type interactions") are occasionally used throughout the manuscript without a sufficient theoretical basis. Clearly defining these concepts at the beginning of the text and using them consistently throughout would improve understanding.

Response 1:  The polarizability density was referenced in the introduction and defined explicitly in equation (3) (section 2) and soft interactions are defined in the first paragraph of the introduction with reference. In order to respond to the criticism of the referee, I have added a brief qualitative description of polarizability density also in the introduction paragraph for sake of completeness and slightly rephrased the definition of soft interactions in the first paragraph.

Comment 2: Some sections, particularly those dealing with QTAIM and Hirshfeld schemes, assume familiarity with advanced partitioning methods. As with the previous comment, including a brief comparative summary or outline would facilitate accessibility for a wider audience.

Response 2: I am surprised by this comment because in section 2 there is a general discussion about partition schemes and a short description of QTAIM, Hirshfeld and Voronoi methods (including a comparison of the domains calculated with the different partitions in Figure 2). Therefore, I cannot see what else could be added.

Comment 3: While, in view of the abstract and the title of the manuscript, a reader might expect to find a discussion of the applications of secondary interactions, particularly for electron donor-acceptor complexes, this is not the case; the content provided focuses mainly on a theoretical comparison of partitioning schemes (QTAIM, Voronoi, and Hirshfeld) and their application to simple homopolar and heteropolar diatomic molecules. This poses some challenges in understanding the manuscript, since the crucial "applications" section is either missing or incomplete; in particular, the use of polarizability density as a predictive tool for secondary interactions is promising but underdeveloped. A more explicit comparison with electrostatic potential-based methods would clarify its added value.

Response 3: I have now added a section on donor-acceptor complexes, discussion a couple of examples, like NH3-I2 and Cr(CO)6, thus encompassing main group and transition metal complexes. 

Comment 4: While the comparison of partitioning schemes is well-executed, the manuscript could benefit from a more in-depth discussion on the practical implications of the observed differences. The paper correctly notes that the atomic polarizabilities depend on the partitioning scheme. But do not specify whether frequency-dependent polarizabilities were considered or whether static values were used exclusively. This distinction is critical for interpreting secondary interaction potentials. The discussion could be enhanced by providing a clear, expert-level perspective on how a computational chemist would choose one method over another for a specific supramolecular system and what the physical meaning of the resulting differences truly is.

Response 4: The calculations of atomic polarizabilities are static. For the interpretation of secondary interactions, within known theories, the high frequency polarizabilities are important. These however are very similar to static polarizabilities, because all low-frequency phenomena (like nuclear rearrangement) are de-activated. As a matter of facts, also in Stone's theory of intermolecular forces, static atomic polarizabilites were used. 

Comment 5: Figure 1 displays the trace of polarizability density, providing a visualisation of the data. Likewise, Figure 3 offers a visual comparison of atomic polarizability ellipsoids for LiF using different schemes. The manuscript could incorporate similar visual representations for secondary interaction applications to illustrate the utility of the distributed atomic polarizability concept. Additionally, the resolution of certain polarizability maps may not be sufficient to distinguish subtle features, especially in polyatomic systems. Higher-quality renderings or zoomed-in panels would be useful. Clear legends and consistent formatting across all figures are also recommended.

Response 5: Two figures were added for coordination complexes and secondary interactions 

Reviewer 4 Report

Comments and Suggestions for Authors

In the manuscripr author discusses methods of partitioning computed molecular polarizability into atomic terms.

Author tested three different partition schemes : two so-called hard-space partitions (QTAIM and Voronoi) and one overlapping densities scheme 
like Hirshfeld. Likewise author presents the polarizability density and distributed atomic polarizabilities for the interpretation of supramolecular interactions.

The above mentioned partition schemes were applied to a set o molecules - diatomics homonuclear and heteronuclear as well as polyatomics.

Results are shown in 8 Figures and 2 Tables and are well discussed.

The value of the manuscript is in presenting the focus on electric polarizability instead of electric potential or charges changes, shiffting thus the perspective from hard-type to soft-type of interactions. 

After carefull reading the manuscript referee points only to few minor issues:

l.630 "CAM-B3LYP,[9]" .. remove comma

l.632 "using couple perturbed Kohn-Sham calculations" ... change to "coupled"

l.627 in "5. Materials and Methods" please mention what basis sets did you use in your CAM-B3LYP calculations

l.638 in "5.2 Elecron" ... fix to "Electron"

l.642 Author claims that the "polarizability density was calculated by numerical differentiation with a locally written routine" Would author be willing to share his program with description+tutorials in github so that interested public can start using this new methodology ? 

Author Response

Commnet 1: l.630 "CAM-B3LYP,[9]" .. remove comma

Response 1: removed

Comment 2: l.632 "using couple perturbed Kohn-Sham calculations" ... change to "coupled"
Response: corrected

Comment 3: l.627 in "5. Materials and Methods" please mention what basis sets did you use in your CAM-B3LYP calculations
Response: I added the basis set (which was previously mentioned only in figure captions and table captions). A reference was also added

Comment 4: l.638 in "5.2 Elecron" ... fix to "Electron"
Response 4: fixed

Comment 5: l.642 Author claims that the "polarizability density was calculated by numerical differentiation with a locally written routine" Would author be willing to share his program with description+tutorials in github so that interested public can start using this new methodology ? 

Response 5: the local routine is now available in github and reference was given.

Reviewer 5 Report

Comments and Suggestions for Authors

In the present mns, Macchi analyzes the methods used to partition the molecular polarizability into atomic terms.

Machi is a well-known scientist in Materials, Theoretical and Inorganic Chemistry.
The mns is well written.
The analysis well presented.

Thus, I recommend the publication of the mns after a minor revision

Minor:
- The fonts at Figures 4-6 should be increased.
- The author uses “we” in many lines. Since only on is the author, instead of using “I”, the author could use passive voice.
-A comment is needed regarding the effect of different methodologies or different functionals on the molecular polarizability.

Author Response

I thank this reviewer for the comments. The referee appreciates the work and has only minor changes that have been taken into account

  • The fonts at Figures 4-6 should be increased.
    • Response: the font sizes was increased 

  • - The author uses “we” in many lines. Since only on is the author, instead of using “I”, the author could use passive voice.
    • Response: Sentences with we have been rephrased accordingly (apart in one case where in fact I refer to a collaborative previous work). These small changes are not highlighted in the resubmitted version

  •  A comment is needed regarding the effect of different methodologies or different functionals on the molecular polarizability.
    • A comment was added, referring to previous studies in which the most appropriate functionals were selected. It was not the purpose of the current work, though. I have added anyway a reference to a study in which various functionals were tested against most accurate post-Hartree Fock calculations.

Round 2

Reviewer 3 Report

Comments and Suggestions for Authors

The manuscript now presents a comprehensive and highly coherent study that effectively addresses the objectives stated in its title and abstract. Several points warrant clarification to further enhance the reader's understanding of the authors' proposals:

The application section now discusses secondary interactions, mainly driven by non-covalent forces. The manuscript should confirm that the chosen computational approach (typically DFT) is suitable for these systems. Authors are encouraged to justify their use of standard functionals and basis sets, or better yet, check select results using a dispersion-corrected DFT (DFT-D3) or higher-level methods like MP2 or CCSD(T). While extensive high-level calculations aren't needed, brief verification ensures trends in distributed polarizability are reliable and not due to incomplete non-covalent energetics.

This work effectively compares QTAIM, Voronoi, and Hirshfeld partitioning. It would benefit from including quantitative metrics, such as correlation coefficients (r²) or average absolute deviations for atomic polarizabilities (αA​) in secondary interaction complexes. These statistics would provide a clearer basis for comparing methods rather than relying on qualitative observation.

A final, comprehensive proofreading by a native English speaker is recommended to address minor instances of awkward phrasing. Further tightening of the Abstract and Conclusion may improve clarity and readability for the wider Molecules audience.

Author Response

Comment #1 The application section now discusses secondary interactions, mainly driven by non-covalent forces. The manuscript should confirm that the chosen computational approach (typically DFT) is suitable for these systems. Authors are encouraged to justify their use of standard functionals and basis sets, or better yet, check select results using a dispersion-corrected DFT (DFT-D3) or higher-level methods like MP2 or CCSD(T). While extensive high-level calculations aren't needed, brief verification ensures trends in distributed polarizability are reliable and not due to incomplete non-covalent energetics.

Response: This issue was already discussed in the previous round of review, raised by another reviewer. As already mentioned, reference 39 discusses the choice of the DFT functional, the performances of which were carefully compared with advanced post-HF methods. Anyway, an additional sentence was now added in the Materials and Methods section.

Comment #2: This work effectively compares QTAIM, Voronoi, and Hirshfeld partitioning. It would benefit from including quantitative metrics, such as correlation coefficients (r²) or average absolute deviations for atomic polarizabilities (αA​) in secondary interaction complexes. These statistics would provide a clearer basis for comparing methods rather than relying on qualitative observation.

Response: A short paragraph concerning averages of absolute discrepancies was added in section 3.1.2

Comment #3: A final, comprehensive proofreading by a native English speaker is recommended to address minor instances of awkward phrasing. Further tightening of the Abstract and Conclusion may improve clarity and readability for the wider Molecules audience.

response: After careful proofreading, minor instances were corrected.